# Conversion of Upper-Limb Inertial Measurement Unit Data to Joint Angles: A Systematic Review

**DOI:** 10.3390/s23146535

**Published:** 2023-07-19

**Authors:** Zhou Fang, Sarah Woodford, Damith Senanayake, David Ackland

**Affiliations:** 1Department of Biomedical Engineering, The University of Melbourne, Melbourne 3052, Australia; zhouf1@student.unimelb.edu.au (Z.F.); swood1@student.unimelb.edu.au (S.W.); damith.senanayake@unimelb.edu.au (D.S.); 2Department of Mechanical Engineering, The University of Melbourne, Melbourne 3052, Australia

**Keywords:** inertial sensors, optoelectronic motion analysis, sensor fusion, glenohumeral joint, wearables, IMU

## Abstract

Inertial measurement units (IMUs) have become the mainstay in human motion evaluation outside of the laboratory; however, quantification of 3-dimensional upper limb motion using IMUs remains challenging. The objective of this systematic review is twofold. Firstly, to evaluate computational methods used to convert IMU data to joint angles in the upper limb, including for the scapulothoracic, humerothoracic, glenohumeral, and elbow joints; and secondly, to quantify the accuracy of these approaches when compared to optoelectronic motion analysis. Fifty-two studies were included. Maximum joint motion measurement accuracy from IMUs was achieved using Euler angle decomposition and Kalman-based filters. This resulted in differences between IMU and optoelectronic motion analysis of $4^{{}^{\circ}}$ across all degrees of freedom of humerothoracic movement. Higher accuracy has been achieved at the elbow joint with functional joint axis calibration tasks and the use of kinematic constraints on gyroscope data, resulting in RMS errors between IMU and optoelectronic motion for flexion–extension as low as $2^{{}^{\circ}}$. For the glenohumeral joint, 3D joint motion has been described with RMS errors of $6^{{}^{\circ}}$ and higher. In contrast, scapulothoracic joint motion tracking yielded RMS errors in excess of $10^{{}^{\circ}}$ in the protraction–retraction and anterior-posterior tilt direction. The findings of this study demonstrate high-quality 3D humerothoracic and elbow joint motion measurement capability using IMUs and underscore the challenges of skin motion artifacts in scapulothoracic and glenohumeral joint motion analysis. Future studies ought to implement functional joint axis calibrations, and IMU-based scapula locators to address skin motion artifacts at the scapula, and explore the use of artificial neural networks and data-driven approaches to directly convert IMU data to joint angles.

## 1. Introduction

Quantification of joint motion has played a key role in our understanding of upper-limb function, from rehabilitation [1,2,3,4], sports science [5,6,7], and ergonomics [8,9,10], to robotics [11,12,13,14]. Joint angles remain the standardized, clinically relevant measure to quantify inter-segment angles at a joint and are critically important for interpreting upper-limb joint function and consolidating data, e.g., across different subjects, laboratories, or motion measurement modalities [15,16,17,18]. Several types of instrumentation have been employed to measure human motion data, including optoelectronic motion analysis [19,20,21], RGB and RGB-D cameras [22,23,24,25], radar [26,27], and ultrasonic measurement devices [28,29]. Optoelectronic motion analysis systems such as Vicon (Oxford Metrics, Oxford, UK) and Optotrak (Northern Digital Inc.,Waterloo, Canada) are considered the gold standard in non-invasive joint-angle measurement, and are used extensively in the evaluation of scapulothoracic, glenohumeral, and elbow joint function [19,30,31]. During movement, video motion analysis systems directly measure 3D trajectories of markers placed on body landmarks at high speed and accuracy, and these data are then used to reconstruct anatomical coordinate systems for the calculation of joint angles between adjacent bones. Unfortunately, these systems are costly, require a dedicated capture space that is typically indoors, are restricted in terms of the available marker capture volume, and are associated with significant setup time before data acquisition. RGB cameras, radar, and ultrasound are susceptible to occlusion between the subject and receiver, and are thus less desirable in a data collection environment with complex or unanticipated object layouts [22,32,33].

We are currently at the frontier of new technological developments in human motion measurement, with commercially available inertial measurement units (IMUs) now inexpensive, lightweight, portable, wireless, and thus highly amenable to “wearable” human motion measurement in and outside of the laboratory environment without limitation on capture volume [34,35]. Modern IMUs can provide orientation data with respect to a local reference system via micro-electromechanical systems (MEMS) comprising tri-axial accelerometers, gyroscopes, and magnetometers. Accelerometers are used to measure the linear acceleration relative to gravity [36,37,38], gyroscopes measure the angular velocity of rotation, and magnetometers provide heading or yaw axis information by measuring the Earth’s magnetic field. Unfortunately, MEMS have hardware limitations that can substantially affect human movement data and sensor usage. For example, accelerometers are sensitive to impact; gyroscopic output, which can be integrated to obtain angular position, is prone to instrumentation noise accumulation resulting in sensor drift; while magnetometers can be sensitive to magnetic disturbances from surrounds [36,39,40,41]. To improve measurement accuracy and reduce orientation estimation errors using IMUs, sensor-fusion algorithms have been developed and are frequently employed, including Kalman-based filters [42,43,44], complementary filters [45,46,47], and gradient descent algorithms [48,49,50]. A recent systematic review by Longo et al. (2022) compared the performance of different sensor-fusion algorithms in the measurement of shoulder joint angles [51]. However, the accuracy of upper-limb joint angles computed using IMUs, which are dependent on the use of sensor-fusion algorithms and the alignment of sensors to anatomical segments, remains poorly understood.

Calculation of joint angles using IMUs is fundamentally different from that using optical motion analysis methods since IMUs cannot be explicitly used to define anatomical landmarks and bony coordinate systems. Instead, a sensor-to-segment calibration is required to establish the angular position relationship between the sensor and the body [36,41,52]. Specifically, IMUs are positioned on the body so that their sensing axes are aligned with anatomical references, such as the longitudinal axis of a bone [30,53]. Static poses and dynamic calibration tasks can also be used to define joint axes of rotation [36,54,55]. However, this requires a well-planned experimental protocol and user experience, and out-of-plane joint motion axes remain challenging to quantify. Another major challenge in the calculation of joint angles using IMUs is skin motion artifacts, which describe the motion of the surface of the skin, in which IMUs are affixed, relative to the underlying bony segments. The scapula, for example, can glide over 10 cm beneath the skin during abduction [20,56].

Several systematic reviews on human upper-limb motion analysis using IMUs have been carried out to date. De Baets et al. (2017) conducted a review of shoulder kinematics measurement using IMUs and showed that protocols for scapulothoracic joint motion quantification demonstrated high reliability and repeatability, while limited consistency was found in humerothoracic joint-angle evaluation. However, these approaches did not perform comparisons relative to a reference motion measurement modality [57]. Other reviews have demonstrated that the accuracy of IMU-based joint-angle measurement is dependent on the specific joint under investigation, the motion task [31,58,59], and is largely driven by the IMU data processing technique employed [34,60,61]. For instance, Walmsley et al. (2018) showed that shoulder joint motion tracking errors using IMUs were lower for single plane movements such as flexion–extension than for multiple degree-of-freedom joint motions. Poitras et al. (2019) investigated the validity and reliability of whole-body movements using IMUs on a joint-by-joint basis, showing that task complexity can increase the variability of out-of-plane shoulder joint angles, including abduction–adduction. Furthermore, five algorithms employed in reconstructing joint motion from IMU data were compared by Filippeschi et al. (2017), with the Kalman filter and QUEST (QUaternion ESTimator) algorithm shown to be the most accurate [34,62]. However, despite numerous studies exploring different sensor processing algorithms across various joints, a consistent approach to the conversion of IMU data to joint angles has not been adopted. The considerable variability and inconsistencies in IMU-derived motion data underscores the need for a standardized modeling approach for IMU to joint-angle conversion.

The aims of this study were two-fold. The first was to evaluate computational methods used to convert IMU data to joint angles in the upper limb, which included the scapulothoracic, humerothoracic, glenohumeral, and elbow joints; the second was to quantify the accuracy of these approaches when compared to optoelectronic motion analysis. The findings will help guide the use of IMUs for upper-limb joint motion measurement in both the research and clinical settings.

## 2. Methods

### 2.1. Database Search Strategy

A literature search was conducted to identify previously published articles that describe the measurement of upper-limb joint angles using IMUs following the PRISMA 2020 protocol for systematic reviews [63]. Articles were identified through a systematic search of the following five databases: Scopus, Web of Science, EMBASE (via Ovid), Medline (via Ovid), and CENTRAL. These databases were searched for English publications before 14 June 2023. To maximize capture of all relevant articles, a broad search strategy was used with the following terms:

IMU* OR inertial measurement unit* OR inertial sensor* OR wearable sensor* OR accelerometer* OR gyroscope* OR magnetometer*

AND

joint angle* OR kinematic* OR range of motion

AND

Upper limb* OR upper extremit* OR shoulder* OR elbow* OR arm* OR humer* OR scapul*

AND

optoelectronic* OR optical OR gold standard OR video* OR camera*

### 2.2. Selection Criteria

After the removal of duplicates from search results, all titles and abstracts were screened using the following inclusion and exclusion criteria (Figure 1).

Inclusion criteria:Motion analysis experiments conducted on human subjectsStudies evaluating joint angles in the upper limb, including those associated with one or more of the shoulder, elbow, and scapula segmentsUse of IMUs that operate with an accelerometer, gyroscope, magnetometer, or a combinationComparison of IMU-based joint angles with those derived from optoelectronic motion analysis.

Exclusion criteria:Non-English studiesThesis, conference papers, or review articlesNon-human studiesStudies that employ sensors other than IMUs

### 2.3. Quality Assessment

The quality of all included studies was evaluated using a customized quality assessment based on the Downs & Black and STROBE checklist [64,65]. The quality assessment questions covered key characteristics of the studies including aim(s), measurement protocols, findings, and error analyses. There were 11 questions in total, and each was scored 0, 1, or 2 which corresponded to not addressed, partially addressed, or fully addressed, respectively. Quality scores were collated, and their mean and range were calculated. High methodological quality was defined as a score of ≥20 (to a maximum of 22), moderate quality was defined as a score of <20 and ≥15, and low quality was defined as a score of <15. Two reviewers participated in the quality assessment independently, and any disagreement in the scores was resolved by discussion. The quality assessment questions included:Is the aim or objective of the study clearly described?Are the main outcomes to be measured clearly defined in the Introduction or Methods section?Are the selection and characteristics of participants included in the study clearly described?Are the details of the experimental setup and measurement procedure clearly described?Are the movement tasks clearly described?Are the kinematics in all degrees of freedom about the joints evaluated?Are the methods of data processing or algorithms used clearly described?Are the findings or key results of the study clearly described?Are the validity and reliability of the experiment described?Are the experimental errors in the results of the studies discussed?Are the limitations and biases of the study discussed?

### 2.4. Data Extraction

For the scapulothoracic, humerothoracic, glenohumeral, and elbow joints, data extracted were summarized by study sample size, sensor-to-segment calibration approach (if any), sensor-fusion algorithm, joint-angle calculation method, motion tasks, and error metrics describing differences between IMU and optoelectronic motion data.

## 3. Results

### 3.1. Search Outcome and Quality

A total of 989 studies were identified from the initial search in the 5 databases, of which 262 duplicates were removed. After title and abstract screening, 136 eligible studies were retrieved for the full-text screening based on the selection criteria. During the full-text screening, two additional studies were included by manually checking the bibliography. Fifty-two studies were included for data extraction (Figure 1). The methodological quality scores of these studies ranged from 9 to 22, with an average score of 17.6. Fourteen studies were of high quality, 32 studies were of moderate quality and 6 studies were of low quality. Studies scored highest on questions related to objectives and outcomes, achieving an average quality score of 2.0. However, quality assessment questions concerning validity and reliability, as well as limitations and bias, received average scores below the 25th percentile (1.3) of the average scores of all studies (Figure 2). Although all included studies validated their methodology against an optoelectronic measurement system, 30 studies did not assess the reliability of their methodology by repeating testing on the same subject or with different operators, leading to a low average score in validity and reliability.

### 3.2. Inertial Measurement Unit (IMU) Placement and Sensor-to-Segment Calibration

In computing upper-limb joint angles, IMUs have been positioned on the torso, scapula, upper arm, and forearm (Table 1). Sensor placement on the body is often chosen to minimize soft tissue or skin motion artifacts and includes the flat portion of the sternum, the lateral distal aspect of the upper arm, and the dorsal distal aspect of the forearm. For the simultaneous collection of IMU and optoelectronic data, retro-reflective markers and IMUs have been placed independently on anatomical landmarks [36,52,60,66,67,68,69,70,71,72,73,74,75,76,77,78,79,80,81,82,83,84,85,86,87,88,89,90], retro-reflective markers placed directly on IMUs [30,41,91,92,93,94,95,96,97,98,99,100,101,102,103,104,105,106], or a combination of both approaches [61,107,108,109,110,111,112] (Figure 3).

To establish a relationship between IMU orientation and that of an underlying anatomical coordinate system, a sensor-to-segment calibration process is typically employed. Common calibration methods include predefined sensor alignment with a segment [66,94,103,104], static pose alignment [67,69,70,71,72,73,74,83,84,88,98,99,100,102,105,109], functional joint movements [92,107], or their combinations [30,36,41,61,68,79,81,91,96,97,106,108,110,112,113,114], as well as use of IMU palpation calipers [90,93] (Figure 4). Predefined sensor alignment involves the placement of an IMU on the body to align with a bone-fixed reference frame. For static pose calibration, a subject may perform one or more pre-determined static postures, for example, standing with the palms facing to the front [107,112], an N-pose with arms neutrally placed alongside the body [41,83,100,106,111], or a T-pose with arms abducted to $90^{{}^{\circ}}$ [69,86,97]. Static calibration then aims to define the sensor coordinate system using gravity as a reference by averaging resultant accelerometer data for a given pose while also establishing a neutral or reference alignment of a joint. In contrast, functional joint movements are performed by repetitively moving a joint through a specific degree of freedom, and anatomical joint axis calculation can be performed using averaged gyroscope data [41,101,106,107,111], or by solving a kinematic constraint function on gyroscope data using optimization [41,75,77]. IMU palpation calipers can be used to identify bony landmark positions for IMU registration [90,93].

### 3.3. Inertial Measurement Unit (IMU) Data to Joint-Angle Conversion

Strategies to reduce gyroscopic drift and magnetic disturbance have been employed on raw IMU data to improve sensor orientation estimates. Slade et al. (2022) corrected gyroscopic drift by keeping IMUs stationary for a period of time, calculating the gyroscopic bias by averaging the angular velocity in each sensing axis, and then eliminating this bias from the raw gyroscopic data [81]. Similar approaches to eliminate gyroscopic drift were applied by Ligorio et al. (2020) with IMUs placed stationary on the floor [41], and Bessone et al. (2019) during a “T-pose” performed by the subject using “aktos-t” software [69]. Truppa et al. (2021) evaluated gyroscopic bias as a variable that was updated during static IMUs data collection. To achieve this, accelerometer data were restricted within a spherical neighborhood of a specific value. At each static frame, the corresponding gyroscopic bias was calculated using sensor coordinate system orthogonalization and then eliminated in subsequent dynamic motions. Magnetic disturbance has also been minimized by sensor calibration in the surrounding magnetic field using spherical [73,95] or ellipsoidal fitting of raw magnetometer data [70]. Additionally, Laidig et al. (2017) proposed a linear model to evaluate heading angle errors in IMU orientation due to magnetic field disturbance, which was solved using optimization and subsequently eliminated [77].

To convert IMU orientation data to joint angles, definitions of anatomical joint coordinate systems have been established using the Denavit–Hartenberg representation [61,67,89], orthonormal [30,41,66,68,70,71,73,79,83,86,91,92,93,94,95,96,97,98,101,106,107,108,109,110], and non-orthonormal segment coordinate systems [36,52,75,113] (Figure 5). The orientation of one segment relative to another is computed using 3D Euler angle decomposition [30,36,41,52,66,68,77,78,79,80,84,85,86,88,90,91,92,93,95,96,97,99,100,106,107,111], or by solving for the planer geometric relationship between two predefined segment vectors [70,73,89,94,109,113]. Other strategies include solving forward kinematics equations established by state-space representation [61,67], or axis-angle-based inverse kinematics [81].

### 3.4. Scapulothoracic Joint Motion Measurement

Three studies, two of high-quality [79,90] and one of moderate quality [30], measured scapular kinematics with respect to the thorax using IMUs [30,79] (Table 2). Cutti et al. (2008) reported RMS errors of between $0.2^{{}^{\circ}}$ and $3.2^{{}^{\circ}}$ for humerothoracic flexion–extension, abduction–adduction, hand-to-nape, hand-to-top-of-head, shoulder girdle elevation–depression and protraction–retraction, when comparing IMU data with optoelectronic motion data via markers placed on IMUs. They defined a set of sensor orientations and positions on the body and adopted static pose calibration. They employed Xsens’ proprietary Kalman filter, followed by Euler angle decomposition, to calculate the 3D scapulothoracic joint angles. This method was also adopted by Parel et al. (2014), who measured scapula kinematics during shoulder flexion and abduction using an optoelectronic-based scapula tracker. This study reported higher errors, especially at high humerothoracic elevation angles (up to $130^{{}^{\circ}}$), which were associated with RMS errors of $10.3^{{}^{\circ}}$ and $11.1^{{}^{\circ}}$ for scapular protraction–retraction and anterior-posterior tilt, respectively. Using the same IMU placement and Euler angle calculation approach, Friesen et al. (2023) incorporated a scapular calibration at two humeral elevation positions using an IMU scapular locator. At maximum humerothoracic elevation during abduction, RMS errors of $12.2^{{}^{\circ}}$, $9.8^{{}^{\circ}}$, and $15.0^{{}^{\circ}}$ were observed for scapulothoracic protraction–retraction, medical-lateral rotation, and anterior-posterior tilt angle, respectively. For a flexion task, RMS errors of$10.8^{{}^{\circ}}$, $9.4^{{}^{\circ}}$, and $18.8^{{}^{\circ}}$ were reported for these three degrees of freedom at the scapulothoracic joint. In the other 8 tasks of daily living, RMS errors for 3D scapulothoracic angles fell within a range of $7.0^{{}^{\circ}}$ to $25.2^{{}^{\circ}}$ at maximum humeral elevation, except for a side reach task which exhibited $43.0^{{}^{\circ}}$, $27.9^{{}^{\circ}}$, and $17.2^{{}^{\circ}}$ scapulothoracic protraction–retraction, mediolateral rotation, and anterior-posterior tilt angle, respectively.

### 3.5. Humerothoracic Joint Motion Measurement

A total of 34 studies measured humerothoracic joint angles using IMUs, which were of high (*n* = 9) [36,41,60,84,88,92,96,100,104], moderate (*n* = 23) [30,61,66,67,68,69,70,72,74,81,85,86,87,93,94,100,102,103,105,106,107,109,115] and low quality (*n* = 2) [73,89] (Table 3). To achieve sensor-to-segment calibration, 6 studies used predefined sensor alignment followed by a static pose [30,61,68,69,81,96], 4 studies combined static calibration with functional joint movements [36,41,91,106], and 11 studies employed static calibration only [67,70,72,73,74,84,88,100,102,105,109]. Twenty studies relied on the manufacturer’s proprietary sensor-fusion algorithm to calculate the sensor orientation [30,36,66,68,69,72,73,74,85,86,87,88,89,93,94,95,100,107,109], 8 studies implemented previously published sensor-fusion algorithms [41,60,70,81,92,96,103,104], while 4 developed a custom sensor-fusion approach [91,102,105,106]. To calculate joint angles from IMUs, 19 studies applied Euler angle decomposition of adjacent segment orientation [30,36,41,66,68,84,85,86,88,91,92,93,95,96,100,102,105,106,107], 5 studies derived joint angles from vector geometry [70,73,89,94,109], 3 studies calculated inclination angle by formula [60,103,104], 2 studies solved established forward kinematics equations [61,67], 3 studies acquired joint angles directly from proprietary software [69,74,87], and 1 study employed axis-angle-based inverse kinematics [81].

Six studies of moderate quality achieved RMS errors or mean absolute errors that were less than $5^{{}^{\circ}}$ in all three degrees of freedom of humerothoracic joint motion, which were of the highest accuracy among the studies on this joint [30,61,91,95,102,105]. Truppa and colleagues (2021) exploited a sensor-fusion algorithm that automatically eliminated gyroscopic bias during a series of yoga poses (mean absolute error < $4^{{}^{\circ}}$) [91]. They mitigated sensor-fusion drift by first defining an orthogonal coordinate system for the thorax and upper arm based on a gravity vector and humeral flexion–extension axis derived from a functional movement calibration. Once static IMU motion was detected with IMUs, the sensor’s local frame was then re-orthogonalized using the gravity vector, and the gyroscope bias was subsequently evaluated and eliminated.

Cutti et al. (2008) obtained RMS errors in the range of 0.2° to 3.2° for humerothoracic flexion–extension, abduction–adduction, internal–external rotation, hand-to-nape, and head-touching using a predefined sensor alignment with a static pose calibration, Xsens Kalman filter and Euler angle decomposition [30].

Zhang et al. (2011) reported RMS errors during arbitrary upper-limb movements of 2.36°, 0.88°, and 2.9° in flexion–extension, abduction–adduction, and internal–external rotation angles, respectively [61]. They defined upper-limb joints as mechanical linkages and modeled angular joint motion as state-space vectors. A neutral pose was performed for sensor-to-segment calibration, and measurement noise at the accelerometer, gyroscope, and magnetometer was modeled as Gaussian white noise with zero mean and finite covariances. An unscented Kalman filter was used to solve forward kinematics equations that related the sensor measurement data to joint angles [116].

Lambrecht et al. (2014) applied a magnetic heading compensation to the InvenSense proprietary sensor-fusion algorithm, which utilized accelerometer and gyroscope data [115]. The raw magnetometer data about the sensor was calibrated by spherical fitting followed by a tilt angle compensation using quaternion output from the sensor fusion. Then, magnetic compensation was used to correct heading angles caused by gyroscopic drift, improving the orientation estimation for long-term data collection. During reaching, RMS errors of 4.9°, 1.2°, and 2.9° were reported for humerothoracic plane, elevation, and axial rotation, respectively.

**Table 3 sensors-23-06535-t003:** Studies that measured humerothoracic joint angles using IMUs, including their sample size, study quality, sensor-to-segment calibration method, sensor-fusion approach, joint-angle calculation method, tasks performed, kinematic errors, and associated error metric when comparing joint angles with those calculated using an optoelectronic motion analysis system. Errors during flexion–extension, abduction–adduction, and internal–external rotation are given in plain text, while errors in the Euler angle plane of elevation, elevation angle, and axial rotation are given in parentheses. Kinematic errors and error ranges [square brackets] are given. Error metrics with “r” represent the right side of the body only. Acronyms used include PSA, predefined sensor alignment; PA, proprietary algorithm; KF, Kalman filter; F/E, flexion/extension; AB/AD, abduction/adduction; IN/EX, internal/external rotation; EAD, Euler angle decomposition; FJM, functional joint movement; MFC, Magnetic field calibration; ABV, angle between vectors.

Study	Sample	Quality Score	Calibration	Sensor Fusion	Joint Angle Calculation	Task	Error Metric	Kinematic Errors		
								F/E (Plane)	AB/AD (Elevation)	IN/EX (Axial Rotation)
[30]	*n* = 1	16	PSA, static	Xsens KF	EAD	Miscellaneous	RMSE	[0.2°, 3.2°]	[0.2°, 3.2°]	[0.2°, 3.2°]
[107]	*n* = 5	19	FJM	Xsens KF	EAD	Miscellaneous	Peak error	(20°)	(10°)	(20°)
[66]	*n* = 1	19	PSA	Xsens KF	EAD	Shoulder F/E	Peak error	13.4°	/	/
						Shoulder horizontal AB/AD		/	17.25°	/
						Shoulder internal rotation		/	/	60.45°
						Water serving	Mean error	13.82°	7.44°	28.88°
[61]	*n* = 4	15	Static	Unscented KF	Forward kinematics	Arbitrary movement	RMSE	2.36°	0.88°	2.9°
[67]	*n* = 8	16	Static	Unscented KF	Forward kinematics	Shoulder F/E	RMSE	5.5°	/	/
						Shoulder AB/AD		/	4.4°	/
[94]	*n* = 1	16	PSA	Xsens KF	ABV	Shoulder F/E	Mean error ± SD	0.76° ± 4.04°	/	/
						Shoulder AB/AD		/	0.69° ± 10.47°	/
						Shoulder IN/EX		/	/	−0.65° ± 5.67°
[95]	*n* = 1	18	PSA	InvenSense PA, MFC	EAD	Reaching	RMSE	(4.9°)	(1.2°)	(2.9°)
[36]	*n* = 10		PSA, static, FJM	Xsens KF	EAD	Shoulder flexion	RMSE± SD	8.0° ± 3.9°	17.8° ± 3.8°	17.5° ± 8°
		21				Shoulder abduction in scapular plane		16.3° ± 4.6°	22.4° ± 3.6°	23.4° ± 6.2°
						Rotating wheel		8.7° ± 2.0°	9.2° ± 3.9°	22.0° ± 10.3°
[92]	*n* = 12	20	FJM	KF	EAD	Miscellaneous	Proportional & Systematic error	0.01X +0.46°	0.21Y +1.3°	0.20Z −0.29°
[96]	*n* = 8	22	PSA, static	Gradient decent	EAD	Front crawl	RMSE	5°	10°	7°
						Breaststroke		/	5°	3°
[102]	*n* = 10	19	Static	PI control	EAD	Shoulder F/E	RMSE	0.63°	1.57°	1.25°
[60]	*n* = 6	21	PSA	Accelerometer	Inclination	Milking	RMSE ± SD	/	(7.2° ± 2.9°)	/
[68]	*n* = 6	19	PSA, static	Xsens KF	EAD	Mimic surgery	RMSE	/	(6.8°)	/
[103]	*n* = 13	19	PSA	Extended KF	Inclination	Move dowels (slow)	RMSE	/	(1.1°± 0.6°)	/
[69]	*n* = 14	16	PSA, static	iSen PA	iSen PA	Shoulder F/E	RMSE	14.6°	/	/
						Shoulder AB/AD		/	10.9°	/
[93]	*n* = 14	16	IMU caliper	Xsens KF	EAD	Arm sagittal plane elevation	RMSE ± SD	/	(4.4° ± 4.1°)	/
						Arm scapular plane elevation		/	(2.5° ± 1.7°)	/
						Arm frontal plane elevation		/	(2.3° ± 2.5°)	/
						Shoulder IN/EX		/	/	(1.8° ± 1.4°)
[104]	*n* = 13	20	PSA	KF	Inclination	Move dowels (slow)	RMSE	/	(1.0°± 0.6°)	/
[105]	*n* = 1	15	Static	ESOQ-2 KF	EAD	Uniaxial arm rotation	RMSE	1.10°	1.42°	1.96°
[41]	*n* = 10	21	Static, FJM, optimization	KF, TRIAD	EAD	Yoga sequence	RMSE	3.4°	7.5°	3.9°
[70]	*n* = 6	14	Static	MFC, gradient decent	ABV	Rowing	% Mean error ± SD (r)	2.19% ± 1.23%	/	/
[109]	*n* = 1	15	Static	Extended KF	ABV	Shoulder AB/AD	RMSE	/	4.7°	/
						Shoulder F/E		5.6°	/	/
[100]	*n* = 11	20	Static	Xsens KF	EAD	Item elevating (easy)	RMSE ± SD	/	(2.18° ± 0.85°)	/
						Item elevating (hard)		/	(2.06° ± 1.23°)	/
[73]	/	10	Static	ADIS16448 PA	ABV	Rowing	Mean absolute error (r)	/	(3.76°)	/
[91]	*n* = 10	18	Static, FJM	Orthogonalization, drift compensation	EAD	Yoga sequence	Mean absolute error	3°	2°	4°
[84]	*n* = 1	21	Static	MyoMotion KF	EAD	Nordic walking	Mean error	−8.2°	−31.7°	/
[85]	*n* = 19	18	Assume aligned	Rebee-Rehab PA	EAD	Flexion	RMSE	7.62°	/	/
						Extension		5.04°	/	/
						Abduction		/	8.75°	/
						External rotation		/	/	10.08°
[72]	*n* = 10	17	Static	Perception Neuron PA	/	Stationary walk	RMSE ± SD	1.9° ± 0.8°	7.14° ± 2.97°	/
						Distance walk		1.12° ± 0.65°	5.36° ± 3.16°	/
						Stationary jog		1.94° ± 1.53°	5.97° ± 3.8°	/
						Distance jog		1.78° ± 1.16°	5.7° ± 2.57°	/
						Stationary ball shot		2.23° ± 1.97°	11.85° ± 10.24°	/
						Moving ball shot		1.99° ± 1.12°	15.15° ± 9.32°	/
[74]	*n* = 15	17	Static	Notch PA	Notch PA	Shoulder AB/AD	Mean error ± SD	/	24.48° ± 4.83°	/
						Shoulder F/E		34.11° ± 3.83°	/	/
						Shoulder IN/EX		/	/	44.95° ± 3.5°
						Hand-to-back pocket		8.7° ± 1.58°	3.05° ± 2.36°	0.1° ± 3.11°
						Hand-to-contralateral shoulder		3.49° ± 1.97°	21.24° ± 4.14°	−1.53° ± 4.75°
						Hand-to-top-of-head		/	21.88° ± 3.1°	14.7° ± 14.13°
[86]	*n* = 24	19	PSA	WaveTrack PA	EAD	Abduction	RMSE	/	12.2°	/
						Adduction		/	12.8°	/
						Horizontal flexion		/	/	13°
						Horizontal extension		/	/	9.7°
						Vertical flexion		14°	/	/
						Vertical extension		17.9°	/	/
						External rotation		/	/	10.7°
						Internal rotation		/	/	10.4°
[87]	*n* = 6	15	PSA	SwiftMotion PA	SwiftMotion PA	Reaching	RMSE	6.82°± 4.33°	/	/
[81]	*n* = 5	19	PSA, static	Mahony filter	Inverse kinematics	Fugl-Meyer task	RMSE ± SD	6.9° ± 4.2°	5.2° ± 0.8°	7.9° ± 2.6°
[106]	*n* = 10	19	Static, FJM	UKF	EAD	Yoga sequence	RMSE	3.2° ± 0.98°	3.85° ± 2.35°	6.90° ± 4.01°
[88]	*n* = 7	20	Static	Perception Neuron PA	EAD	Flexion	RMSE	9.2°	/	/
						Extension		3.4°	/	/
						Adduction		/	7.6°	/
						Abduction		/	11.4°	/
						Internal rotation		/	/	7.4°
						External rotation		/	/	8.1°
						Box lifting		8.8°	6.8°	8.2°
[89]	*n* = 1	12	Regression modelling	gForcePro+ PA	ABV	Grasping	RMSE	6.3°	4.1°	6.5°

Madrigal et al. (2016) applied a proportional-integral (PI) control algorithm to fuse gyroscope- and accelerometer-based estimations of a single IMU orientation on the upper arm, and then used Euler angle decomposition [117]. For upper-arm flexion to 90°, they achieved an RMS error of 0.63°, 1.57° and 1.25° in humerothoracic flexion–extension, abduction–adduction, and internal–external rotation, respectively. Duan et al. (2020) obtained a similar accuracy of 1.10°, 1.42°, and 1.96° for roll, pitch, and yaw angles of a single IMU placed on the upper arm during uniaxial arm rotations. They combined the Second Estimator of the Optimal Quaternion (ESOQ-2) [118,119] with a Kalman filter to calculate sensor orientation.

Perez et al. (2010) attached IMUs to a subject via a garment and assumed a fixed sensor-to-segment orientation. However, due to the sliding of the garment relative to the skin, shoulder internal–external rotation movements resulted in motion errors of over $60^{{}^{\circ}}$.

### 3.6. Glenohumeral Joint Motion Measurement

Six studies that included 1 of high quality [97] and 5 of moderate quality [71,98,99,108,110], measured glenohumeral joint angles using IMUs. Five of these studies used an Xsens IMU system and the Xsens-defined biomechanical model known as the MVN model, which consists of 23 segments and 22 joints, for kinematic analysis [71,97,98,108,110] (Table 4). All studies used proprietary sensor-fusion algorithms, while 4 studies relied on proprietary software to compute the joint angles.

Robert-Lachaine et al. (2017) achieved the highest joint kinematics accuracy among included studies (RMS error $\leq 3^{{}^{\circ}}$). ISB definitions of joint coordinate systems were employed for both IMU and optoelectronic systems by calculating rotation matrices that transferred marker clusters fixed with IMUs to bony landmarks using the optoelectronic measurement system. Euler angle decomposition was subsequently used to calculate glenohumeral joint angles during a box-moving task [97]. Pedro et al. (2021) also obtained good accuracy in glenohumeral joint-angle measurement during tennis forehand drives using Xsens software (RMS error $\leq 6.1^{{}^{\circ}}$). This was achieved using an MVN model for both IMU and optoelectronic systems [110].

### 3.7. Elbow Joint Motion Measurement

A total of 39 studies of high (*n* = 9) [36,41,52,84,88,92,96,97,113], moderate (*n* = 25) [30,61,66,67,68,69,70,71,72,74,80,81,83,93,94,98,99,101,106,108,110,111,112,115] and low quality (*n* = 5) [73,77,78,82,89] compared elbow joint angles derived from IMUs with those from optoelectronic systems. Twenty-six of these studies measured motion about the two primary degrees of freedom of the elbow joint, flexion–extension and pronation–supination. Twelve studies performed a sensor calibration using static poses only [67,70,71,72,73,74,78,83,84,88,98,99], while 13 studies combined static calibration with functional joint movement calibration [41,91,96,97,106,108,110,111,112,113] (Table 5). A total of 21 studies implemented Kalman or Kalman-based filters for sensor-fusion algorithms [30,36,41,61,66,67,68,71,77,80,83,84,92,93,94,97,98,101,106,108,110], 4 studies exploited gradient descent-based algorithm [70,78,96,113], while 11 studies relied on proprietary algorithms to obtain the senor orientation [69,72,73,74,82,88,89,95,99,111,112].

Ten studies reported RMS errors of $\leq 5^{{}^{\circ}}$ for elbow flexion–extension and pronation–supination using IMUs [30,41,52,80,83,89,91,92,93,111], while 5 studies reported RMS errors of $\leq 5^{{}^{\circ}}$ in the flexion–extension direction only [72,73,94,108,110]. The highest accuracy for both degrees of freedom was achieved by Laidig et al. (2022), who applied a kinematic constraint to gyroscopic data to solve for joint axes in each sensor coordinate system using Gauss–Newton optimization algorithm. By combining the optimized joint axes with a novel magnetometer-free sensor-fusion algorithm called 6D VQF algorithm [120], they achieved for pick-and-place and drinking tasks a mean RMS error of 2.1° and 3.7° in elbow flexion–extension and pronation–supination, respectively.

Muller et al. (2016) applied a similar kinematic constraint to gyroscope data for the evaluation of the joint axis, which was solved using the Moore–Penrose pseudoinverse. In door-opening tasks, RMS errors were 2.7° and 3.8° in flexion–extension and pronation–supination, respectively [80]. Ligorio et al. (2017) proposed a four-step functional calibration method that involved planar forearm and upper-arm movements, achieving RMS errors that were less than 4° during both elbow flexion–extension and pronation–supination tasks [111]. For the same movement tasks, Picerno et al. (2019) obtained comparable accuracy with a novel sensor-calibration method that employed a customized IMU caliper device to identify bony landmarks, thus allowing the definition of an anatomical coordinate system [93].

Mavor et al. (2020) obtained mean RMS errors of approximately 40° for the pronation–supination angle of the left and right elbow during 8 military movements [98]. They used the Xsens MVN model in the calculation of IMU angles, while a biomechanical model based on anatomical landmarks was used in optoelectronic motion analysis (Visual 3D). Humadi (2021) measured elbow flexion–extension angles during box-moving, box-elevation, and reaching tasks using the MVN model for IMUs and ISB coordinates systems for optoelectronic motion analysis [71]. An offset error of up to 26° was found in the IMU-based joint angles which predominantly contributed to a total RMS error of approximately 30°.

## 4. Discussion

The purpose of this systematic review was to assess strategies for upper-limb joint-angle calculation using IMUs and their accuracy when compared to optoelectronic motion analysis. Due to skin motion artifacts and challenges associated with tracking dynamic scapula motion, the accuracy of IMU-based joint-angle calculations is generally highest at the humerothoracic and elbow joints and lowest at the scapulothoracic joint and glenohumeral joints. Although scapular landmarks can be digitized using an optoelectronic system, this cannot be achieved using IMUs, and consequently, most upper-limb motion studies using inertial sensors focus on the measurement of humerothoracic motion. The use of Euler angle decomposition resulted in the highest accuracy of humerothoracic, glenohumeral, and elbow joint-angle measurements using IMUs; however, joint-angle calculations are strongly dependent on the sensor-fusion approach employed.

Humerothoracic motion measurement using IMUs is a convenient approach to quantifying upper-limb motion since it does not require measurement and modeling of scapular motion. This has included assessment of upper-limb range of motion and mobility, and sports performance, including real-time applications [60,96,121,122]. Alignment of IMUs with respect to the thorax and humeral segments can be achieved using calibration approaches such as static poses and dynamic functional tasks, which facilitate the establishment of anatomical coordinate systems. For example, Truppa et al. (2021) used accelerometer data during a static standing task to determine a vertical axis, and orthogonal projections of gyroscope data during upper-arm flexion–extension to define a lateral axis, resulted in high joint motion accuracy [91]. Such calibration approaches ensure greater consistency in joint axis definitions with anatomical landmark-based optoelectronic motion analysis and avoid alignment errors that can be introduced from manual sensor placement. High humerothoracic motion analysis accuracy was also achieved by computing orientations of anatomical segments using a sensor-fusion algorithm such as a Kalman-based filter [30,61,105]. This approach is capable of predicting and updating sensor orientation on a recursive basis, taking sensor noise into account, and enabling noise reduction and robust sensor orientation estimation [51]. However, the accuracy of MEMS data fusion is generally dependent on an undistorted surrounding magnetic field. Magnetic field calibration can account for potential magnetic field disturbances, or magnetometer data can simply be omitted from the sensor fusion. This may reduce the sensor’s ability to accurately evaluate the heading angle [52,81,120].

Measurement of scapulothoracic angles using IMUs can be challenging since the scapula slides considerably under the skin during upper-limb elevation, and fixing sensors to the scapula is difficult to achieve [123]. Conventional methods for measuring dynamic scapular motion using an optoelectronic system have involved the use of a scapular tracker [124] or an acromial marker cluster [123,125,126]. These approaches enable anatomical landmarks on the scapula to be digitized across a small number of postures and mapped to a scapular-fixed marker cluster using a regression model. This ultimately facilitates the estimation of scapula movement during continuous dynamic upper-limb motions. Van den Noort et al. (2015) developed an IMU scapula locator to register the alignment of the scapula at different humeral elevations, thus allowing the measurement of scapular motion during upper-limb tasks [127]. Friesen et al. (2023) adopted this approach and employed a regression model to facilitate the interpolation of scapular angles between bone positions registered with the IMU scapula locator [90].

At the glenohumeral joint, the highest accuracy in joint motion measurement using IMUs was achieved by Pedro et al. (2021) using Xsens’ proprietary software [110], which involved the use of an MVN model to establish segment coordinate systems [128]. The tracking of the scapula used a skin-placed IMU, static poses, and functional joint movements to align sensor coordinate systems with the scapula. Joint angles were subsequently computed using Euler angle decomposition between the scapula and humerus. Motion measurement at the glenohumeral joint is substantially affected by skin motion artifacts at the scapula, and segment orientation accuracy is dependent on sensor-fusion algorithm performance. Future development of fast and efficient scapular location methods, including the use of IMU-based scapula locators, will improve glenohumeral joint-angle measurement accuracy.

Elbow joint motion measurement using IMUs has been shown to produce more accurate joint angles than those associated with humerothoracic or glenohumeral joint motion since this motion is more constrained and less influenced by skin motion artifacts than other upper-limb joints. The highest elbow joint motion accuracy was achieved with the use of functional joint movements [41,111] and by applying kinematic constraints to elbow joint axes using optimization [41,52,75]. Once elbow joint axes were established, flexion–extension was typically selected as the first axis of the Euler angle decomposition, which minimized propagation of IMU signal error through the Euler sequence and resulted in optimal flexion–extension motion accuracy [129]. The limited range of elbow carrying angle motion was associated with reduced variability in out-of-plane movements measured during uniaxial elbow flexion–extension motion, as well as reduced joint-angle crosstalk relative to multi-degree-of-freedom joints such as the shoulder [31,59].

Several limitations of this study ought to be considered. First, there was variability in the way joint angles were computed using optoelectronic motion analysis, including retro-reflective marker placement i.e., on landmarks, over body suits, or directly to IMUs, as well as the joint coordinate system definitions, which may ultimately make direct comparisons between studies more subjective. Second, the IMU accuracy metrics were not consistent across studies, with some adopting RMS error, peak error and mean absolute error, which can make interpreting accuracy across studies challenging. Third, a variety of IMU models were employed, each with different sample rates, sensitivity, and fidelity, which may affect joint-angle predictions. Finally, this study focused on activities of daily living, and the results may be different for high-speed joint motions, which were generally not considered in the studies considered, including those during throwing, swimming, and impact sports.

As machine-learning and artificial-intelligence (AI) approaches to data analytics evolve, these techniques are likely to have a greater role in advancing human motion analysis using IMUs. Artificial neural networks have been used to analyze large datasets of IMU sensor data, identify human movement patterns and generate joint angles in an automated manner [130,131,132]. For example, Senanayake et al. (2021) developed a generative adversarial network (GAN) that predicted 3D ankle joint angles using raw IMU data, achieving an accuracy of 3.8°, 2.1° and 3.5° in dorsiflexion, inversion, and axial rotation, respectively [133]. Mundt et al. (2020) estimated 3D lower limb joint angles during gait using a feedforward neural network and achieved RMS errors lower than 4.8°, with the best results in the sagittal motion plane [134]. These findings indicate the feasibility of accurate and reliable computation of joint angles using data-driven approaches without dependence on conventional sensor-fusion algorithms such as Kalman filters. Such models, once trained, can also operate in real time, and provide robust motion analysis that is not dependent on accurate sensor placement. However, the performance of these approaches depends on sufficient quantity and diversity of training data, which may not always be practical to obtain. An inadequate training dataset may result in limited generalizability and model robustness to new subjects and different onboard MEMS hardware configurations, and an inability to predict new pathological movements [134,135,136,137]. Furthermore, the generation of artificial neural networks such as GANs can be challenging due to the large number of hyperparameters that require tuning, which has limited their uptake to date. Thus, the usability and accessibility of these models is also a challenge that must be addressed.

With the increasing availability of low-cost wearable technology, and the establishment of robust joint-angle calculation methods, greater applications of IMUs will be realized, including remote real-time monitoring and telemedicine, particularly in the elderly and motor-compromised [138,139,140], sports training, and human performance optimization [141,142,143], defense applications such as measurement and monitoring of front-line soldiers [98,144], habitual motion evaluation over extended periods including in submarines and spaceflight, and film and animation applications [145,146].

## 5. Conclusions

This systematic review evaluated the accuracy of IMU to joint-angle conversion methods in the upper limb. Due to challenges associated with tracking dynamic scapula motion, motion measurement accuracy using IMUs is generally higher at the humerothoracic and elbow joints, and lowest at the scapulothoracic joint and glenohumeral joints. For humerothoracic and elbow joint motion measurement, maximum measurement accuracy was achieved using sensor-fusion algorithms that include Kalman-based filters to integrate accelerometer, gyroscope, and magnetometer data, and Euler angle decomposition of adjacent IMU-based segment orientations. Optimization-based kinematic constraints on gyroscope data, together with functional joint movement calibration, were also employed for the estimation of elbow joint axes, leading to high-accuracy elbow joint-angle calculation. Future approaches to calculating upper-limb joint angles using IMUs ought to leverage static or functional calibration tasks to establish joint axes of rotation for Euler angle decomposition, implement fast and user-friendly scapula locator jigs to aid in scapulothoracic and glenohumeral joint motion measurement, and draw on robust AI-based algorithms for robust, real-time IMU to joint-angle conversion.

## Figures and Tables

**Figure 1 sensors-23-06535-f001:**
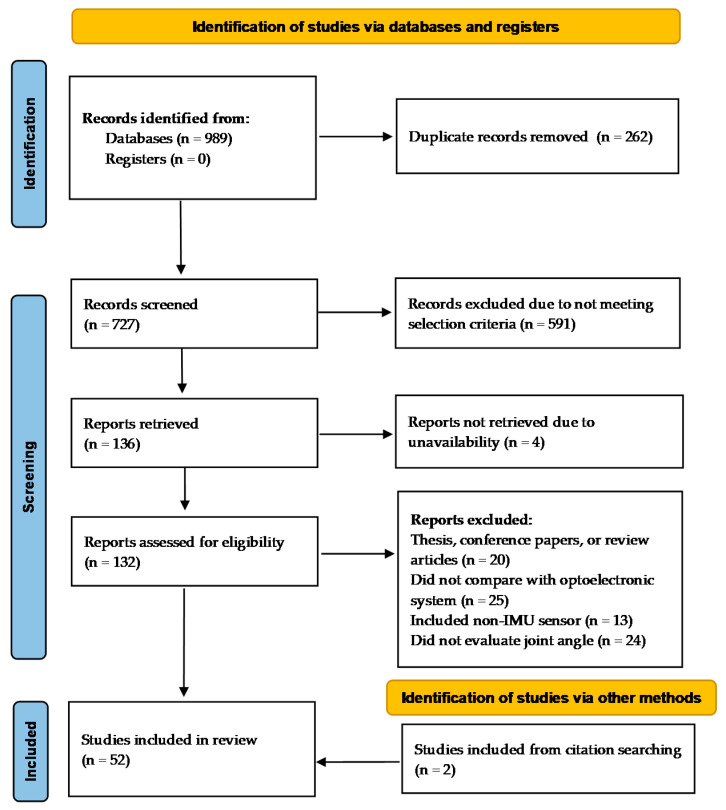
PRISMA flow diagram for systematic review.

**Figure 2 sensors-23-06535-f002:**
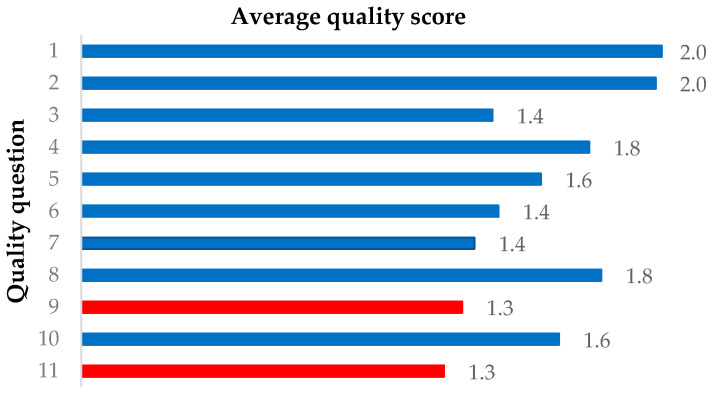
Average quality score for each quality assessment question, rounded to 1 decimal place. The questions with an average score below the 25th percentile, 1.3, are highlighted in red.

**Figure 3 sensors-23-06535-f003:**
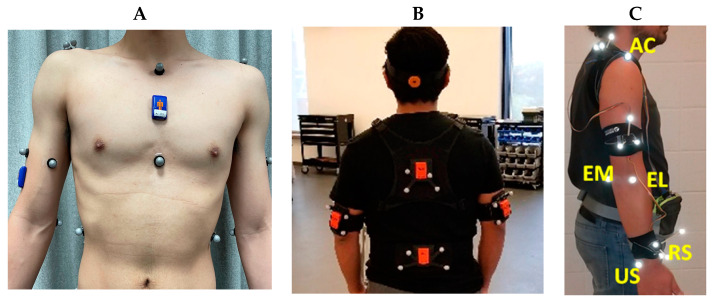
Retro-reflective marker and IMU placement including (**A**) independent marker and IMU placement on anatomical landmarks (**B**) retro-reflective marker cluster attachment directly to IMUs, and (**C**) retro-reflective marker placement on IMUs and directly to anatomical landmarks. Subfigure B adapted from [100] with permission from Human Kinetics, Inc. Subfigure C adapted from [111] with permission from Elsevier.

**Figure 4 sensors-23-06535-f004:**
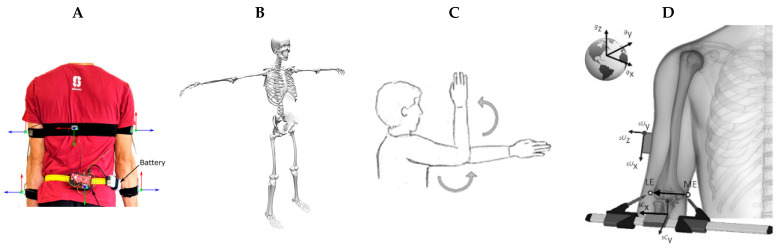
IMU sensor-to-body calibration methods, including (**A**) predefined sensor alignment (**B**) static pose (T-pose) (**C**) functional joint movements, and (**D**) use of an IMU palpation caliper. Subfigure A adapted from [81] with permission from IEEE, Subfigure C adapted from [92] with permission from Elsevier, and Subfigure D adapted from [93] with permission from Nature Portfolio.

**Figure 5 sensors-23-06535-f005:**
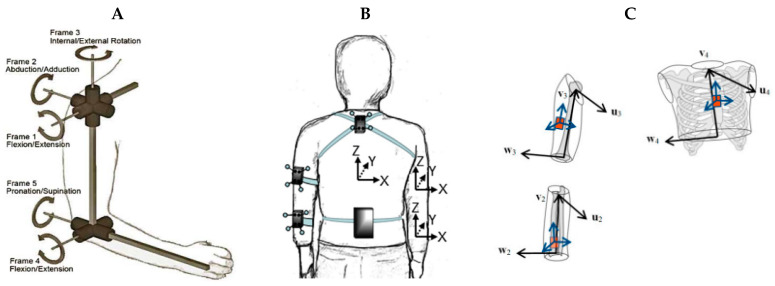
Strategies to define anatomical coordinate systems using IMUs including (**A**) Denavit–Hartenberg joint representation (**B**) orthonormal segment coordinate system (**C**) non-orthonormal segment coordinate system. Subfigure A was adapted from [61] with permission from IEEE, Subfigure B was adapted from [92] with permission from Elsevier, and Subfigure C was adapted from [36] with permission from MPDI.

**Table 1 sensors-23-06535-t001:** Placement of IMUs used to measure scapulothoracic, humerothoracic, glenohumeral, or elbow joint angles in all included studies. Acronyms used include ST, scapulothoracic joint; HT, humerothoracic joint; GH, glenohumeral joint; EL, elbow joint.

Study	Reported Joint Angle	IMU Placement Position			
		Torso	Scapula/Shoulder	Upper Arm/Humerus	Forearm
[101]	EL	/	/	Lateral, distal upper arm	Dorsal, distal forearm
[30]	ST, HT, EL	Sternum	Cranial, central-third scapular spine	Central-third, lateral-posterior upper arm	Dorsal, distal forearm
[107]	HT	Sternum	/	Lateral, distal upper arm	Dorsal, distal forearm
[66]	HT, EL	Middle back	/	Along external triceps long head	Dorsal, distal forearm
[61]	HT, EL	Sternum	/	Lateral, distal upper arm	Dorsal, distal forearm
[67]	HT, EL	/	/	Lateral, middle upper arm	Dorsal, distal forearm
[94]	HT, EL	Central, frontal trunk	/	Lateral, middle upper arm	Dorsal, distal forearm
[95]	HT, EL	Sternum	/	Upper arm	Distal forearm
[79]	ST	Sternum	Cranial, central-third scapular spine	Central-third lateral-posterior upper arm	/
[36]	HT, EL	Sternum	/	Central-third Lateral, upper arm	Dorsal, distal forearm
[92]	HT, EL	Central back, below neck	/	Middle, lateral-posterior upper arm	Middle, dorsal-posterior forearm
[96]	HT, EL	Sternum	/	Central-third, lateral-posterior upper arm	Dorsal, distal forearm
[76]	EL	/	/	Lateral upper arm, bony region	Dorsal, distal forearm
[102]	HT	/	/	Posterior, distal upper arm	/
[75]	EL		/	Distal upper arm	Distal forearm
[60]	HT	Sternal notch	/	Lateral, middle upper arm	
[77]	EL	/	/	Distal upper arm	Distal forearm
[111]	EL	/	/	Lateral, middle upper arm	Dorsal, distal
[68]	HT, EL	Sternum	/	Lateral, middle upper arm	Dorsal, middle forearm
[97]	GH, EL	Sternum	Scapula	Lateral, distal upper arm	Dorsal, distal
[103]	HT	/	/	Lateral, middle upper arm	/
[69]	HT, EL	Sternum	/	Lateral, middle upper arm	Dorsal, middle forearm
[93]	HT, EL	Sternum	/	Lateral, distal upper arm	Dorsal, distal forearm
[104]	HT	/	/	Lateral, middle upper arm	/
[105]	HT	/	/	Lateral, middle upper arm	/
[41]	HT, EL	Middle sternum	/	Lateral, middle upper arm	Dorsal, middle forearm
[70]	HT, EL	Central back	/	Lateral, middle upper arm	Dorsal, middle forearm
[109]	HT	Central back	/	Lateral, middle upper arm	Dorsal, middle forearm
[98]	GH, EL	Sternum	Acromion	Lateral, middle upper arm	Dorsal, middle forearm
[99]	GH, EL	Sternum	Scapula	Lateral, distal upper arm	Dorsal, distal forearm
[108]	GH, EL	Sternum	Mid scapular spine	Lateral, middle upper arm	Dorsal, middle forearm
[78]	EL	Central, frontal trunk	/	Lateral, middle upper arm	Dorsal, distal forearm
[100]	HT	Central back	/	Distal, lateral-posterior upper arm	/
[113]	EL	/	/	Lateral, middle upper arm	Dorsal, middle forearm
[71]	GH, EL	Sternum	Acromion	Upper arm	Forearm
[110]	GH, EL	Sternum	Scapula	Lateral, distal upper arm	Dorsal, distal forearm
[73]	HT, EL	Central back	/	Lateral, middle upper arm	Dorsal, middle forearm
[91]	HT, EL	Sternum	/	Upper arm	Forearm
[83]	EL	/	/	Lateral, lower 1/3 upper arm	Dorsal, lower 1/3 forearm
[84]	HT, EL	C7 vertebrae	/	Lateral, middle upper arm	Dorsal, distal forearm
[85]	HT	/	/	/	Dorsal, middle forearm
[72]	HT, EL	Central back, below neck	Scapular superior angle	Lateral, middle upper arm	Dorsal, distal forearm
[74]	HT, EL	Central, frontal trunk	/	Anterior, middle upper arm	Radial, middle forearm
[86]	HT	Sternum	/	Anterior, middle upper arm	/
[52]	EL	/	/	Distal upper arm	Distal forearm
[112]	EL	/	/	Lateral, middle upper arm	Dorsal, middle forearm
[87]	HT	T2 vertebrae	/	Lateral, distal upper arm	Dorsal, distal forearm
[81]	HT, EL	Central back	/	Lateral, middle upper arm	Dorsal, middle forearm
[106]	HT, EL	Sternum	/	Lateral upper arm	Lateral forearm
[88]	HT, EL	T8 vertebrae	Cranial scapula	Lateral, distal upper arm	Dorsal, distal forearm
[89]	HT, EL	/	/	Lateral, middle upper arm	Dorsal, middle forearm
[90]	ST	Sternum	Acromion/mid-scapular spine	Posterior, distal upper arm	/

**Table 2 sensors-23-06535-t002:** Studies that measured scapulothoracic joint angles using IMUs, including their sample size, study quality, sensor-to-segment calibration method, sensor-fusion approach, joint-angle calculation method, tasks performed, kinematic errors and associated error metric when comparing joint angles with those calculated using an optoelectronic motion analysis system. Kinematic errors and error ranges [square brackets] are given. Acronyms used include PSA, predefined sensor alignment; KF, Kalman filter; F/E, flexion/extension; AB/AD, abduction/adduction; EAD, Euler angle decomposition.

Study	Sample	Quality Score	Calibration	Sensor Fusion	Joint Angle Calculation	Task	Error Metric	Kinematic Errors		
								Protraction-Retraction	Medial- Lateral Rotation	Anterior-Posterior Tilt
[30]	*n* = 1	16	PSA, static	Xsens KF	EAD	Miscellaneous	RMSE	[0.2°, 3.2°]	[0.2°, 3.2°]	[0.2°, 3.2°]
[79]	*n* = 23	20	PSA, static	Custom	EAD	Shoulder F/E	Peak RMSE	10.3°	5°	11.1°
						Shoulder AB/AD		7.1°	5°	7.5°
[90]	*n* = 30	21	PSA, IMU scapula locator	Xsens KF	EAD	Abduction	RMSE at maximum humeral elevation	12.2°	9.8°	15°
						Flexion		10.8°	9.4°	18.8°
						Comb hair		9.9°	7°	14.9°
						Wash axilla		10.8°	13.4°	20.2°
						Tie apron		12°	13.7°	25.2°
						Over head reach		13.4°	11.8°	14.1°
						Side reach		43°	27.9°	17.2°
						Forward transfer		14.1°	13.3°	17.4°
						Floor lift		13.6°	15.8°	13.9°
						Overhead lift		17.9°	12.8°	14.7°

**Table 4 sensors-23-06535-t004:** Studies that measured glenohumeral joint angles using IMUs, including their sample size, study quality, sensor-to-segment calibration method, sensor-fusion approach, joint-angle calculation method, tasks performed, kinematic errors, and associated error metric when comparing joint angles with those calculated using an optoelectronic motion analysis system. Error metrics with “r” representing the right side of the body only. Acronyms used include PSA, predefined sensor alignment; PA, proprietary algorithm; KF, Kalman filter; F/E, flexion/extension; AB/AD, abduction/adduction; IN/EX, internal/external rotation; EAD, Euler angle decomposition; FJM, functional joint movement.

Study	Sample	Quality Score	Calibration	Sensor fusion	Joint Angle Calculation	Task	Error Metric	Kinematic Errors		
								F/E	AB/AD	IN/EX
[97]	*n* = 12	20	Static, FJM	Xsens KF	EAD	Box moving	RMSE	35.8°	19.7°	40.2°
[98]	*n* = 10	19	Static	Xsens KF	Xsens PA	Military movements	RMSE ± SD (r)	19.1° ± 15°	15.2° ± 8.75°	31.0°± 26.0°
[99]	*n* = 5	19	Static	Perception Neuron PA	EAD	Box moving	RMSE	17.5°	10.9°	16°
[108]	*n* = 10	18	Static, FJM	Xsens KF	Xsens PA	Gymnastics move	RMSE	12.57°	9.86°	8.46°
[71]	*n* = 10	18	Static	Xsens KF	Xsens PA	Box moving	RMSE (r)	12.3°	6.7°	33.8°
						Box elevation		14.6°	6.9°	29°
						Reaching at head height		15.8°	7.8°	31.7°
[110]	*n* = 29	18	Static, FJM	Xsens KF	Xsens PA	Tennis ball hitting	RMSE	6.1°	3.5°	4.1°

**Table 5 sensors-23-06535-t005:** Studies that measured elbow joint angles using IMUs, including their sample size, study quality, sensor-to-segment calibration method, sensor-fusion approach, joint-angle calculation method, tasks performed, kinematic errors, and associated error metric when comparing joint angles with those calculated using an optoelectronic motion analysis system. Kinematic errors and error ranges [square brackets] are given. Error metrics with “r” represent the right side of the body only. Acronyms used include PSA, predefined sensor alignment; PA, proprietary algorithm; KF, Kalman filter; F/E, flexion/extension; P/S, Pronation/supination; IN/EX, internal/external rotation; EAD, Euler angle decomposition; FJM, functional joint movement; MFC, Magnetic field calibration; ABV, angle between vectors.

Study	Sample	Quality Score	Calibration	Sensor Fusion	Joint Angle Calculation	Task	Error Metric	Kinematic Errors	
								F/E	P/S
[101]	*n* = 1	16	FJM	KF	Rotation matrix, least square filter	Eating routine	RMSE	21°	/
						Grooming routine		7°	/
[30]	*n* = 1	16	PSA, static	Xsens KF	EAD	Elbow F/E and P/S	RMSE	[0.2°, 3.2°]	[0.2°, 3.2°]
[66]	*n* = 1	19	PSA	Xsens KF	EAD	Elbow flexion and P/S	Peak error	5.8°	24.1°
						Water serving	Mean error	18.6°	11.7°
[61]	*n* = 4	15	PSA, static	Unscented KF	Forward kinematics	Arbitrary movement	RMSE	6.2°	13.0°
[67]	*n* = 8	16	Static	Unscented KF	Forward kinematics	Elbow F/E	RMSE	6.5°	/
						Elbow P/S		/	5.5°
[94]	*n* = 1	16	PSA	Xsens KF	ABV	Elbow F/E	Mean error ± SD	−0.54° ± 2.63°	/
						Elbow P/S		/	−5.16° ± 4.5°
[95]	*n* = 1	18	PSA	InvenSense PA, MFC	EAD	Reaching	RMSE	7.9°	1.5°
[36]	*n* = 10	21	PSA, static, FJM	Xsens KF	EAD	Elbow F/E	RMSE ± SD	18.7° ± 2.7°	/
						Elbow P/S		/	15.8° ± 6.3°
						Rotating wheel		20.0° ± 3.7°	/
[92]	*n* = 12	20	FJM	KF	EAD	Miscellaneous	Proportional & Systematic error	0.00X +2.00°	−0.00Z −1.20°
[96]	*n* = 8	22	PSA, static	Gradient decent	EAD	Simulated front crawl	RMSE	15°	10°
						Simulated breaststroke		8°	6°
[82]	*n* = 3	9	/	Invensense PA	INMOCAP PA	Elbow F/E	%RMSE	2.44%	/
[75]	*n* = 1	18	Static, auto-calibration	Xsens KF	Kinematic constraint, EAD	Door opening	RMSE	2.7°	3.8°
[77]	*n* =1	13	Joint axis optimization	Xsens KF, MFC	Kinematic constraint, EAD	Pick-and-place,	Mean error ± SD	4.09° ± 3.43°	−5.16° ± 6.63°
						drinking			
[111]	*n* = 15	18	FJM, static	YEI PA	EAD	Elbow F/E	RMSE	4°	/
						Elbow P/S		/	4°
[68]	*n* = 6	19	PSA, static	Xsens KF	EAD	Mimic surgery	RMSE	8.2° ± 2.8°	/
[97]	*n* = 12	20	Static, FJM	Xsens KF	EAD	Box moving	RMSE	6.2°	12.2°
[69]	*n* = 14	16	PSA, static	iSen PA	iSen PA	Elbow F/E	RMSE	27.1°	/
[93]	*n* = 14	16	IMU caliper	Xsens KF	EAD	Elbow F/E	RMSE	1.9° ± 2.6°	/
						Elbow P/S		/	2.9° ± 1.6
[41]	*n* = 10	21	Static, FJM, optimization	KF, TRIAD	EAD	Yoga sequence	RMSE	3°	3.3°
[70]	*n* = 6	14	Static	MFC, gradient decent	ABV	Simulated rowing	% Mean error ± SD r	2.19% ± 1.23%	/
[98]	*n* = 10	19	Static	Xsens KF	Xsens PA	Military movements	RMSE ± SD r	10.9° ± 5.3°	40.5° ± 27.6°
[99]	*n* = 5	19	Static	Perception Neuron PA	EAD	Box moving	RMSE	14.9°	14.3°
[108]	*n* = 10	18	Static, FJM	Xsens KF	Xsens PA	Gymnastics move	RMSE	4.2°	/
[78]	*n* = 10	10	Static	Madgwick filter	Euler angle	Walking	%RMSE (r)	5.80%	/
[113]	*n* = 1	21	Static, MFC, FJM	Madgwick filter	ABV	Elbow F/E	RMSE (r)	8.23°	/
						Elbow F/E with P/S		9.36°	/
						Walking		5.98°	/
						Simulated front crawl		5.6°	/
						Simulated rowing		6.53°	/
[71]	*n* = 10	18	Static	Xsens KF	Xsens PA	Box moving	RMSE (r)	28.2	/
						Box elevation		30.7	/
						Reaching at head height		34.2	/
[110]	*n* = 29	18	Static, functional	Xsens KF	Xsens PA	Tennis ball hitting	RMSE	1.5°	13.1°
[73]	/	10	Static	ADIS16448 PA	ABV	Rowing	Mean absolute error (r)	3.28°	/
[91]	*n* = 10	18	Static, functional	Orthogonalization, drift compensation	EAD	Yoga sequence	Mean absolute error	2°	4°
[83]	*n* = 1	17	Static	KF	Rotation about fixed axis	Elbow F/E	RMSE	3.82°	/
						Elbow P/S		/	3.46°
[84]	*n* = 1	21	Static	KF	EAD	Nordic walking	Mean error (r)	23.7°	/
[72]	*n* = 10	17	Static	Perception Neuron PA	/	Stationary walk	RMSE ± SD	3.4° ± 2.15°	/
						Distance walk		2.04° ± 1.48°	/
						Stationary jog		3.89° ± 2.96°	/
						Distance jog		1.92° ± 1.0°	/
						Stationary ball shot		2.81° ± 2.18°	/
						Moving ball shot		3.2° ± 1.75°	/
[52]	*n* = 2	20	Kinematic constraint, optimization	6D VQF	EAD	Pick-and-place, drinking	RMSE	2.1°	3.7°
[112]	*n* = 15	18	Static, FJM	Notch PA	Notch PA	Tennis hitting	RMSE	5.76°	6.66°
[74]	*n* = 15	17	Static pose	Notch PA	Notch PA	Elbow F/E	Mean error ± SD	17.55° ± 3.28°	/
						Hand-to-contralateral-shoulder		9.91° ± 3.18°	/
						Hand-to-top-of-head		3.34° ± 3.48°	/
[81]	*n* = 5	19	PSA, static	Mahony filter	Inverse kinematics	Fugl-Meyer task	RMSE	5.2° ± 2.1°	/
[106]	*n* = 10	19	Static, FJM	Unscented KF	EAD	Yoga sequence	RMSE	2.96° ± 0.95°	6.79° ± 2.31°
[88]	*n* = 7	20	Static	Perception Neuron PA	EAD	Flexion	RMSE	8.7°	/
						Extension		5.8°	/
						Pronation			7.2°
						Supination		/	7.8°
						Box lifting		12.5°	9.5°
[89]	*n* = 1	12	Regression modelling	gForcePro+ PA	ABV	Grasping	RMSE	3.4°	3.9°

## Data Availability

The data presented in this study are available on request from the corresponding author.

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
