# Peer review of "Conversion of Upper-Limb Inertial Measurement Unit Data to Joint Angles: A Systematic Review"

_sensors, 2023, doi:10.3390/s23146535_

Round 1

Reviewer 1 Report

1. Please number the table in the paper.

2. In the paper, some sentences' Xsens' have been written as' Xsense ', please revise them uniformly.

3. Suggest objectively discussing the advantages and disadvantages of artificial intelligence technology in the field of human motion analysis in the paper.

Author Response

Please see attached response

Reviewer 2 Report

1. Manuscript doesnot have a conclusion section. A separate section with this title must be included summarizing quantitative research findings and future scope of this work. 

2. Update the list of references from 2022 and 2023. Discuss some of them in introduction and also consider for comparison

3.  Can the last two paras of introduction can be titled respectively as key research gaps and contributions ?.. Bullet points can be used. 

4. What is the use of Table 4?

Language is generally ok

Author Response

See attached response

Reviewer 3 Report

Paper title:  Conversion of upper limb inertial measurement unit data to joint angles: A systematic review

Specific comments:

1- What is the main question addressed by the research? The motivation for the study should be further emphasized, particularly; the main advantages of the results in the paper comparing with others should be clearly demonstrated.

2- Do you consider the topic original or relevant in the field? Does it address a specific gap in the field? The research gap is also missing.

3- What does it add to the subject area compared with other published material? The comparison with the existing studies is not sufficient.

4- What specific improvements should the authors consider regarding the methodology? What further controls should be considered? The modification is not explicitly explained point by point. The mathematical proof to explain the improvement that has been noticed in the proposed method is also missing and mandatory.

5- Are the conclusions consistent with the evidence and arguments presented and do they address the main question posed? The abstract is not cleanly written as well as the conclusion. Analysis of the results is missing in the paper. There is a big gap between the results and conclusion.

6- Are the references appropriate? More references on recent work should be added as the number of references is low (maybe you could include more references from recent years:

7- Please include any additional comments on the tables and figures.

There are some points need to be further clarified:

A. The motivation for the study should be further emphasized, particularly; the main advantages of the results in the paper comparing with others should be clearly demonstrated.

B. The example section needs to be further expanded and include some remarks to show the effectiveness and efficiency of the proposed method, compared with others.

C. Some remarks on the main results would be necessary and helpful.

D. The limitations of the studies work should be added.

E. The research gap also is missing.

Minor editing of English language required

Author Response

See attached response

Reviewer 4 Report

This systematic review evaluates computational methods for converting IMU data to joint angles and compares their accuracy to optoelectronic motion analysis. Forty studies were analyzed, revealing that Euler angle decomposition and Kalman-based filters achieved the highest measurement accuracy for IMUs. The results from different works are well-documented. The contributions of the work are significant and interesting for the researchers working in the domain of upper limb movement estimation. There are a few major concerns that need to be addressed before publication:

1. The Introuction should be updated to include the motivation of this review work. More works where joint angle estimation is useful, should be cited such as rehabilitation devices, exoskeleton robots, occupational therapy, etc.

2. In the selection criteria, the authors should justify why they have not considered the thesis, conference papers, and review papers. In literature, There are some good quality conference papers for the joint angle estimation of upper-limb.

3. In view of comment 2, its is surprising that the authors have committed the point (ii) of exclusion criteria from PRISMA diagram. The discrepancy should be checked and revised.

4. To improve the readability, the authors are suggested to present a small separate section on different methods of joint angle estimation using IMU sensors. 

5. Moreover, there should be more information (another section) on the placement and calibration methods of IMU sensors on the upper limb.,as this information is crucial to address. 

6. The limitations of the topic and future research opportunities should be highlighted carefully in the Discussion section to boost the quality of the review work.

Author Response

See attached response

Round 2

Reviewer 3 Report

the authors are respond to all my concerns

 Minor editing of English language required

Reviewer 4 Report

The manuscript's quality and contributions have significantly improved after the revisions. Therefore, the manuscript could be accepted without further revisions.